# Memory-Efficient Visual Autoregressive Modeling with Scale-Aware KV Cache Compression

**Kunjun Li**[♣,◇]    **Zigeng Chen**[◇]    **Cheng-Yen Yang**[♣]    **Jenq-Neng Hwang**[♣]

University of Washington[♣]   National University of Singapore[◇]

{kunjun, zigeng99}@u.nus.edu, {cycyang, hwang}@uw.edu

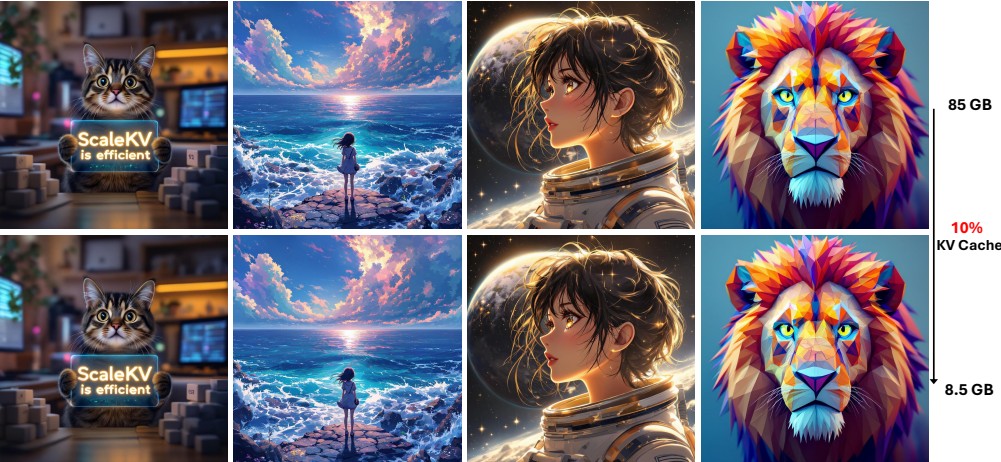

Figure 1: We introduce a new KV cache compression framework for Visual Autoregressive modeling that preserves pixel-level fidelity. On Infinity-8B, it achieves **10x** memory reduction from **85 GB** to **8.5 GB** with negligible quality degradation (GenEval score **remains** at **0.79** and DPG score marginally decreases from **86.61** to **86.49**).

## Abstract

Visual Autoregressive (VAR) modeling has garnered significant attention for its innovative next-scale prediction approach, which yields substantial improvements in efficiency, scalability, and zero-shot generalization. Nevertheless, the coarse-to-fine methodology inherent in VAR results in exponential growth of the KV cache during inference, causing considerable memory consumption and computational redundancy. To address these bottlenecks, we introduce ScaleKV, a novel KV cache compression framework tailored for VAR architectures. ScaleKV leverages two critical observations: varying cache demands across transformer layers and distinct attention patterns at different scales. Based on these insights, ScaleKV categorizes transformer layers into two functional groups: drafters and refiners. Drafters exhibit dispersed attention across multiple scales, thereby requiring greater cache capacity. Conversely, refiners focus attention on the current token map to process local details, consequently necessitating substantially reduced cache capacity. ScaleKV optimizes the multi-scale inference pipeline by identifying scale-specific drafters and refiners, facilitating differentiated cache management tailored to each scale. Evaluation on the state-of-the-art text-to-image VAR model family, Infinity, demonstrates that our approach effectively reduces the required KV cache memory to **10%** while preserving **pixel-level fidelity**. Code is available at https://github.com/StargazerX0/ScaleKV.

39th Conference on Neural Information Processing Systems (NeurIPS 2025).

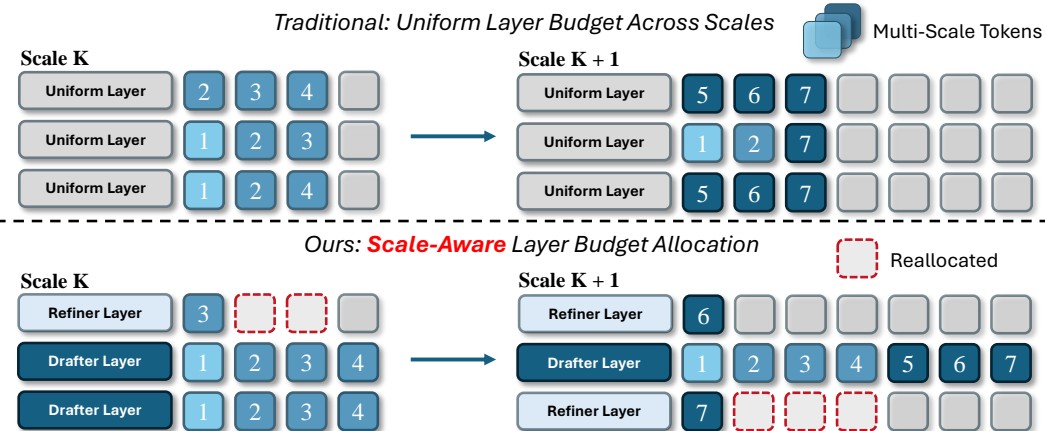

Figure 2: By implementing scale-aware layer budget allocation, ScaleKV enables differentiated cache management tailored to each layer's computational demands at every scale.

# 1 Introduction

Recent advances in Autoregressive (AR) models [51, 23, 68, 31] have achieved impressive image quality and multi-modal capabilities [34, 63, 61, 58, 60] for unified vision understanding and generation. However, the token-by-token generation approach requires numerous decoding steps. Visual Autoregressive (VAR) modeling [53] has revolutionized this process through next-scale prediction, enabling models to decode multiple tokens in parallel. Building upon this framework, several approaches [16, 52] have demonstrated promising results for VAR-based text-to-image generation.

Despite these advancements, VAR models face a fundamental scalability challenge due to exponential growth in token sequence across scales. Unlike traditional next-token prediction models, which process one token per step with linear KV cache growth, VAR models must preserve KV states from all previous token maps. Generating a 1024×1024 image requires processing over 10K tokens across multiple scales, creating severe memory bottlenecks—with KV cache alone consuming approximately 85 GB of memory when generating 1024×1024 images with a batch size of 8 using Infinity-8B [16], a text-to-image VAR model. Figure 3(a) illustrates this complexity, where each scale contributes a new KV cache entry of size $h_k \times w_k$. The total cache requirement grows cubically with the number of scales $n$, while the computational complexity of attention reaches $\mathcal{O}(n^4)$. These memory constraints and increased inference latency significantly impede practical deployment.

To address these inefficiencies, we analyze the specific properties of VAR's next-scale prediction paradigm. First, we observe that cache demands vary significantly across transformer layers. Next, we find that different scales exhibit distinct attention patterns. These findings indicate that VAR requires both layer-adaptive and scale-specific cache management strategies for optimal performance.

**Our Approach.** Inspired by these observations, we propose *Scale-Aware KV Cache* (ScaleKV), a simple yet highly effective method that significantly reduces inference memory while maintaining high generation quality. Our approach categorizes transformer layers into two functional groups: **Drafters** and **Refiners**. Drafters distribute attention across multiple scales to access global information from preceding tokens, requiring greater cache capacity. In contrast, refiners focus attention on current token map to process local details, necessitating substantially reduced cache storage. As illustrated in Figure 2, ScaleKV optimizes multi-scale inference by identifying scale-specific drafters and refiners, facilitating differentiated cache management to their computational demands at each scale.

Extensive evaluation demonstrates the effectiveness of our method. As shown in Figure 1, compared to the original Infinity-8B model, ScaleKV achieves negligible quality degradation (GenEval score remains at 0.79 and DPG score decreases slightly from 86.61 to 86.49) while requiring merely **10%** of the original GPU memory consumption. These results validate that ScaleKV effectively addresses the fundamental memory bottlenecks that have constrained the practical deployment of VAR models.

In conclusion, we introduce ScaleKV, a novel KV cache compression framework for VAR. ScaleKV categorizes transformer layers into drafters and refiners and implements scale-aware layer budget

allocation. Through extensive experiments, our method achieves significant memory reduction while preserving pixel-level fidelity, enabling efficient deployment in resource-constrained environments.

## 2 Related Works

**Autoregressive Visual Generation.** Early works [6, 54] pioneered pixel-by-pixel image generation, later enhanced by VQVAE [55] and VQGAN [8] through image patch quantization. Recent advances include GPT-style models [51, 34], mixture-of-experts [23], linear attention [26], diffusion-autoregressive hybrids [63, 81, 14], and masked approaches [26, 4, 38]. However, autoregressive approaches suffer from substantial inference latency due to sequential token generation. VAR [53] overcomes this limitation through hierarchical parallel decoding, and has been extended to text-to-image synthesis [16, 74, 52, 40, 27, 31], audio synthesis [45], and 3D content creation [73].

**Efficient Visual Generation.** For diffusion models, efficiency optimization methods are already well-developed. [47, 71, 37, 48, 69] focus on reducing sampling steps while [29, 80, 9, 72, 67, 28, 49, 24] optimize models through quantization [24], pruning [9] or knowledge distillation [19]. Several approaches [41, 59, 76, 66, 50, 25, 39, 79] skip redundant computations during the denoising process.

However, research on memory optimization for VAR image generation remains in its early stages. LiteVAR [64] and FastVAR [15] enhance inference speed but do not address fundamental memory bottlenecks, while CoDe [7] improves memory efficiency through collaborative decoding but requires an additional VAR model. Hack [44] reduces memory via per-head budgets, but its irregular tensor shapes require specialized kernels. In contrast, our approach directly reduces memory consumption and integrates with existing techniques to enhance efficiency.

**KV Cache Compression.** Current KV cache compression techniques for large language models (LLMs) and vision-language models (VLMs) primarily utilize quantization [36, 70, 22, 17], eviction [78, 35, 42, 46, 30], and merging [77, 33, 56, 57] strategies. Quantization reduces precision but faces granularity limitations; eviction removes less important tokens using attention metrics while optimizing allocation within budgets [12, 65, 10, 11]; and merging consolidates redundant KV pairs.

However, primarily designed for single-sequence processing in LLMs and VLMs, existing techniques fail to accommodate the multi-scale operational characteristics of VAR and the varying attention patterns across layers and scales.

## 3 Methods

### 3.1 Preliminary

Visual Autoregressive modeling [53] advances traditional AR approaches by shifting the prediction paradigm from "next token" to "next scale." Within this framework, each autoregressive operation produces a token map corresponding to a specific resolution scale, rather than individual tokens.

Given an image feature map $f \in \mathbb{R}^{h \times w \times C}$, VAR quantizes it into $K$ multi-scale token maps $R = (r_1, r_2, \ldots, r_K)$ with increasingly finer resolutions. The joint probability distribution over these multi-scale maps is decomposed autoregressively according to:

$$p(r_1, r_2, \ldots, r_K) = \prod_{k=1}^{K} p(r_k \mid r_1, r_2, \ldots, r_{k-1}), \tag{1}$$

where each token map $r_k \in \{1, \ldots, V\}^{h_k \times w_k}$ contains $h_k \times w_k$ discrete tokens, selected from a vocabulary of size $V$ at scale $k$. At each autoregressive step $k$, the model generates all $h_k \times w_k$ tokens comprising $r_k$ in parallel, conditioning on previously generated scales $(r_1, \ldots, r_{k-1})$. While VAR provides substantial improvements in both inference efficiency and generation quality, this coarse-to-fine generation strategy significantly expands the sequence length.

### 3.2 Key Observations

VAR represents an innovative paradigm that diverges from traditional autoregressive approaches. In this work, we examine the next-scale prediction process to identify properties that can be leveraged to reduce computational redundancy. Our analysis focuses specifically on attention patterns in VAR.

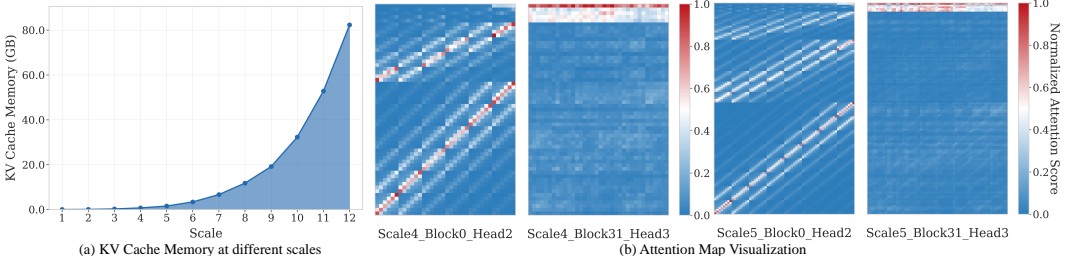

Figure 3: (a) Exponential KV cache growth. (b) Visualization of two distinct attention patterns.

**Cache demands vary significantly across transformer layers.** Our analysis revealed two distinct attention pattern typologies in VAR models, visualized in Figure 3(b). The left pattern (Block0_Head2) exhibits a diagonal structure with dispersed attention that spans preceding scales, indicating broad contextual integration and high cache demand. In contrast, the right pattern (Block31_Head3) demonstrates highly concentrated attention focused predominantly within the current token map, suggesting localized processing with minimal cache requirements. Through systematic analysis of these attention maps, we identified that most layers fall into one of these two categories, which we term **Drafters** and **Refiners**. Drafters distribute attention broadly across historical context, necessitating substantial KV cache capacity to access multiple scales. Refiners, however, concentrate attention primarily on local information within the current token map, requiring significantly reduced cache storage.

**Different scales exhibit distinct attention patterns.** As VAR progresses through its hierarchy, both groups display scale-dependent evolution. Drafter layers exhibit increasingly dispersed attention patterns at higher scales to integrate broader contextual information. Conversely, refiners grow progressively more concentrated, as evidenced by the heightened focus in Scale5_Block31_Head3 compared to its Scale4 counterpart (Figure 3(b)). This bidirectional evolution reveals a specialized hierarchical process where drafters gather global context while refiners perform localized processing.

These findings challenge both uniform cache allocation [62] and position-based cache reduction [3] employed by current methods, suggesting that VAR models would benefit from adaptive allocation strategies accounting for both layer-specific requirements and scale-dependent characteristics.

### 3.3 Scale-Aware KV Cache

Based on our observations, we propose a simple yet highly effective KV cache compression framework for next-scale prediction called Scale-Aware KV Cache. As illustrated in Figure 4, ScaleKV categorizes transformer layers into two functional groups termed **Drafters** and **Refiners**, implementing adaptive cache management strategies based on these roles. This approach optimizes multi-scale inference by identifying each layer's function at every scale, enabling adaptive cache allocation that aligns with specific computational demands of each layer.

**Identifying Drafter and Refiner Layers.** To systematically distinguish between drafter and refiner layers across different scales, we introduce the **Attention Selectivity Index (ASI)**. This metric quantifies each layer's attention patterns by considering two critical factors: (1) the proportion of attention directed to current token map and (2) the concentration of attention in history sequence.

Let $\alpha_{i,j}^{(l,k)}$ represent the normalized attention score from query position $i$ in the current map $r_k$ to key/value position $j$ in layer $l$ when processing scale $k$. The token indices can be partitioned into history indices $\mathcal{P}_{k-1}$ (from previously generated maps $r_1, \ldots, r_{k-1}$) and current map indices $\mathcal{C}_k$ (from $r_k$). The ASI for layer $l$ at scale $k$ is defined as:

$$ASI^{(l,k)} = \underbrace{\mathbb{E}_{i \sim \mathcal{U}(H_k)}\left[\sum \alpha_{i,j}^{(l,k)}, \, j \in \mathcal{C}_k\right]}_{\text{Current Attention Ratio}} \cdot \underbrace{\mathbb{E}_{i \sim \mathcal{U}(H_k)}\left[\text{TopKSum}'\big(\{\alpha_{i,j}^{(l,k)} \mid j \in \mathcal{P}_{k-1}\}\big)\right]}_{\text{History Top-K Ratio}}, \quad (2)$$

where $\mathbb{E}_{i \sim \mathcal{U}(H_k)}[\cdot]$ denotes expectation over query positions $i$ sampled uniformly from the current map $r_k$, and $\text{TopKSum}'(\cdot)$ computes the sum of the top-$K'$ attention scores directed toward the history tokens. The parameter $K'$ controls the number of top scores included in the selectivity term.

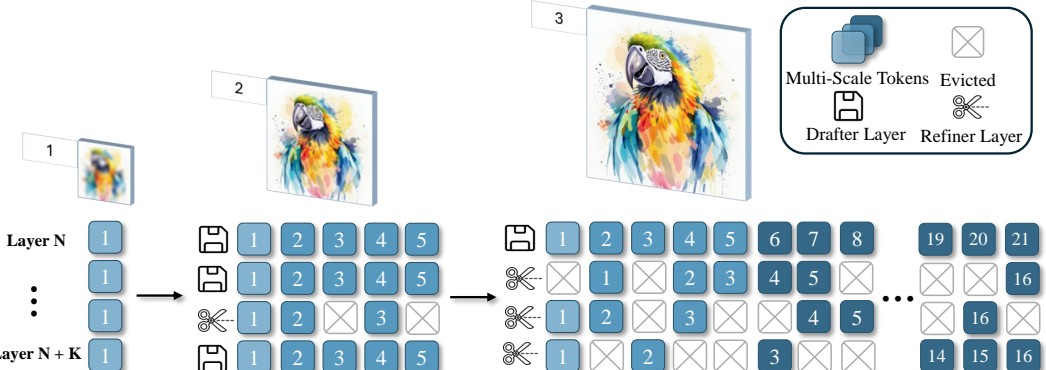

Figure 4: Overview of ScaleKV. Our method categorizes transformer layers into drafters (require extensive cache for global context) or refiners (process local details with minimal cache). This scale-wise identification enables adaptive cache allocation based on each layer's computational demands.

Intuitively, a high $ASI^{(l,k)}$ value indicates that the layer either focuses strongly on the current map or exhibits high selectivity toward specific history tokens, or both. This suggests the layer is functioning as a refiner. Conversely, a low $ASI^{(l,k)}$ value indicates the layer distributes attention more broadly across the prefix context, characteristic of drafter behavior.

Since raw $ASI^{(l,k)}$ values vary significantly across scales due to differences in token counts and attention patterns, we normalize these values within each scale using Z-scores. Let $\mathcal{S} = \{(l, k) \mid 1 \leq l \leq L, 1 \leq k \leq K\}$ be the set of all layer-scale pairs. We rank these pairs by their Z-scores and define the set of drafters $\mathcal{D}$ as the $N_d$ pairs with the lowest Z-scores:

$$\mathcal{D} = \{(l, k) \in \mathcal{S} \mid Z^{(l,k)} \leq Z_{(N_d)}\}, \qquad Z^{(l,k)} = \frac{ASI^{(l,k)} - \mu_k}{\sigma_k + \epsilon}, \tag{3}$$

where $Z_{(N_d)}$ represents the $N_d$-th smallest Z-score. The remaining constitute the refiners $\mathcal{R} = \mathcal{S} \setminus \mathcal{D}$.

The identification process occurs prior to inference using minimal calibration data. Our experiments demonstrate that a set of 10 prompts is sufficient to accurately determine the drafters and refiners.

**Cache Budget Allocation.** After identifying drafters and refiners, we establish an efficient budget allocation strategy that satisfies the same total memory consumption as uniform budget allocation $B_{\text{uniform}}$ while implementing a scale-dependent reduction for refiners:

$$\sum_{k=1}^{K} (N_d^k \cdot B_d(k) + N_r^k \cdot B_r(k)) = B_{\text{uniform}} \cdot L \cdot K, \qquad B_r(k) = B_r(0) - \delta \cdot k. \tag{4}$$

Here, $N_d^k$ and $N_r^k$ represent drafter/refiner layer counts at scale $k$, while $B_d(k)$ and $B_r(k)$ denote their respective cache budgets. The parameter $\delta$ controls the refiner budget decay rate. By leveraging the second observation that refiner attention exhibits increasing concentration at higher scales, refiner cache budgets are linearly reduced from the initial refiner budget $B_r(0)$ as scale $k$ increases. The saved memory is subsequently reallocated to drafters, ensuring $B_d(k) \gg B_r(k)$ to align with the scale-specific computational demands of each layer.

**KV Cache Selection.** After establishing cache budgets for drafter and refiner layers, we implement an efficient token selection strategy to determine which specific KV states should be preserved.

For each token map $r_k$, we first partition the map into $N$ patches and select the centroid token from each patch to form an observation window $\mathcal{W}$. This sampling approach ensures spatial coverage across the token map while maintaining a minimal memory footprint. We then evaluate the relative importance of the remaining tokens based on their attention interactions with the observation window.

Similar to [30], for each attention head $h$, we compute an importance score $s_i^h$ for each token $i$ by measuring the cumulative attention it receives from the observation window tokens:

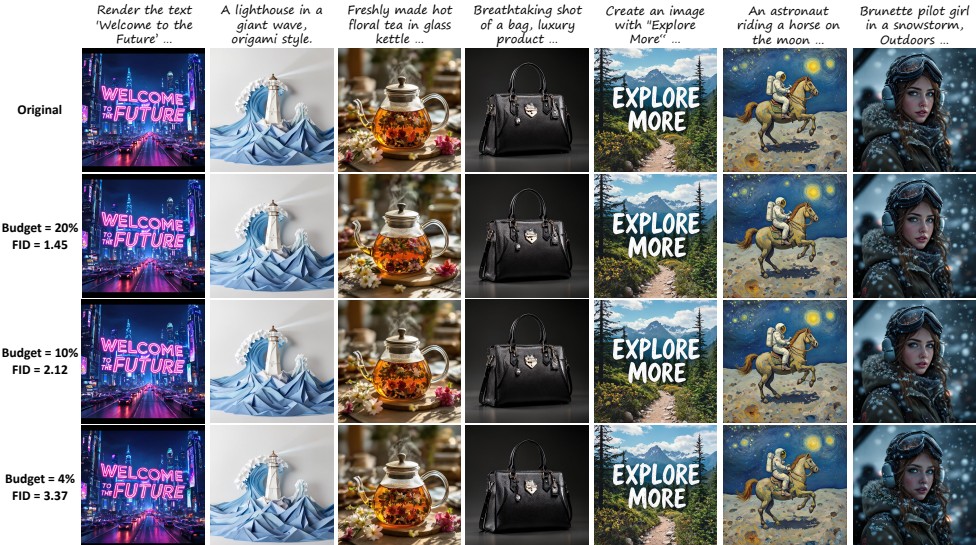

Figure 5: Qualitative comparison between the original Infinity-8B model and our proposed ScaleKV.

$$s_i^h = \sum_{j \in \mathcal{W}} A_{ji}^h, \qquad A^h = \mathrm{softmax}(Q^h \cdot (K^h)^\top / \sqrt{d_k}). \tag{5}$$

Here, $A_{ji}^h$ represents the normalized attention weight from query token $j$ in the observation window to key token $i$, and $d_k$ denotes the dimension of the key vectors. This formulation effectively quantifies each token's contribution to the contextual representation of the observation window.

After calculating cumulative attention, we aggregate these scores through average pooling to produce a unified importance metric. For each layer $l$, we then select the top-$k^l$ tokens with the highest aggregated importance scores, where $k^l$ corresponds to the layer-specific cache budget determined by our allocation strategy. Only the KV states of these selected tokens, along with the observation window tokens, are preserved in the cache for subsequent steps, while all others are efficiently pruned.

The observation window is compact (typically comprising only 16 tokens), enabling efficient attention score computation between this window and the remaining tokens with negligible computational overhead. Rather than hindering performance, our experimental results show this approach yields notable speedups of up to 1.25× through significantly reduced KV cache memory requirements.

## 4 Experiments

### 4.1 Experimental Setup

**Base Models**. We evaluated ScaleKV on two VAR-based text-to-image models of different capacities: Infinity-2B and Infinity-8B [16], to validate our method's generalizability across model scales. To represent practical deployment scenarios with varying resource constraints, we analyzed performance under three memory budget constraints: 4%, 10%, and 20% of the original KV cache size.

**Dataset and Metrics**. We assessed output consistency with the original models using the MS-COCO 2017 [32] validation set, which comprises 5,000 images and captions. Consistency was measured by the Fréchet Inception Distance (FID) [18], Learned Perceptual Image Patch Similarity (LPIPS) [75], and Peak Signal-to-Noise Ratio (PSNR). We also used two established image generation benchmarks GenEval [13] and DPG [20] for perceptual quality and semantic alignment with input prompts. For memory efficiency, we report the KV cache memory usage measured with a batch size of 8. We use GPT-4o [21] to generate 10 prompts for calibrating drafters and refiners. Our drafter/refiner identification method is effective across different calibration prompt sources, with results converging after analyzing only a small sample of prompts, as detailed in the appendix.

Table 1: Quantitative comparisons of output consistency on MS-COCO 2017 dataset.

| Method | Budget | Infinity-2B | | | | Infinity-8B | | | |
|---|---|---|---|---|---|---|---|---|---|
| | | KV Cache | FID ↓ | LPIPS ↓ | PSNR ↑ | KV Cache | FID ↓ | LPIPS ↓ | PSNR ↑ |
| Full Cache | 100% | 38550 MB | - | - | - | 84328 MB | - | - | - |
| Sliding Window [1] | 20% | 7800 MB | 5.63 | 0.17 | 20.71 | 17062 MB | 4.82 | 0.14 | 20.99 |
| StreamingLLM [62] | 20% | 7800 MB | 3.85 | 0.12 | 22.00 | 17062 MB | 3.94 | 0.14 | 21.65 |
| SnapKV [30] | 20% | 7800 MB | 3.25 | 0.12 | 22.51 | 17062 MB | 3.10 | 0.10 | 22.65 |
| PyramidKV [3] | 20% | 7800 MB | 3.23 | 0.11 | 22.62 | 17062 MB | 3.03 | 0.10 | 22.76 |
| **ScaleKV** | 20% | 7800 MB | **1.82** | **0.08** | **24.84** | 17062 MB | **1.45** | **0.06** | **25.60** |
| Sliding Window [1] | 10% | 3900 MB | 8.58 | 0.24 | 18.99 | 8531 MB | 8.71 | 0.20 | 19.02 |
| StreamingLLM [62] | 10% | 3900 MB | 5.49 | 0.19 | 19.79 | 8531 MB | 6.29 | 0.17 | 19.97 |
| SnapKV [30] | 10% | 3900 MB | 4.66 | 0.16 | 20.83 | 8531 MB | 4.68 | 0.15 | 20.60 |
| PyramidKV [3] | 10% | 3900 MB | 4.52 | 0.16 | 20.92 | 8531 MB | 4.69 | 0.14 | 20.79 |
| **ScaleKV** | 10% | 3900 MB | **2.53** | **0.11** | **22.64** | 8531 MB | **2.12** | **0.09** | **23.25** |
| Sliding Window [1] | 4% | 1590 MB | 16.68 | 0.30 | 17.49 | 3478 MB | 19.23 | 0.27 | 17.50 |
| StreamingLLM [62] | 4% | 1590 MB | 8.71 | 0.25 | 18.31 | 3478 MB | 8.54 | 0.22 | 18.63 |
| SnapKV [30] | 4% | 1590 MB | 5.10 | 0.24 | 18.23 | 3478 MB | 6.68 | 0.19 | 19.15 |
| PyramidKV [3] | 4% | 1590 MB | 5.51 | 0.23 | 18.65 | 3478 MB | 6.55 | 0.19 | 19.26 |
| **ScaleKV** | 4% | 1590 MB | **3.51** | **0.16** | **20.82** | 3478 MB | **3.37** | **0.12** | **21.41** |

Table 2: Quantitative comparisons of perceptual quality on GenEval and DPG Benchmarks.

| Methods | # Params | GenEval | | | | DPG | | |
|---|---|---|---|---|---|---|---|---|
| | | Two Obj. | Position | Color Attri. | Overall ↑ | Global | Relation | Overall ↑ |
| SDXL [43] | 2.6B | 0.74 | 0.15 | 0.23 | 0.55 | 83.27 | 86.76 | 74.65 |
| LlamaGen [51] | 0.8B | 0.34 | 0.07 | 0.04 | 0.32 | - | - | 65.16 |
| Show-o [63] | 1.3B | 0.80 | 0.31 | 0.50 | 0.68 | - | - | 67.48 |
| PixArt-Sigma [5] | 0.6B | 0.62 | 0.14 | 0.27 | 0.55 | 86.89 | 86.59 | 80.54 |
| HART [52] | 0.7B | 0.62 | 0.13 | 0.18 | 0.51 | - | - | 80.89 |
| DALL-E 3 [2] | - | - | - | - | 0.67 | 90.97 | 90.58 | 83.50 |
| Emu3 [58] | 8.5B | 0.81 | 0.49 | 0.45 | 0.66 | - | - | 81.60 |
| Infinity-2B [16] | 2.0B | 0.84 | 0.43 | 0.57 | **0.725** | 89.01 | 90.03 | **83.06** |
| **+ ScaleKV (10%)** | 2.0B | 0.84 | 0.44 | 0.55 | **0.730** | 82.45 | 90.48 | **83.01** |
| Infinity-8B [16] | 8.0B | 0.89 | 0.61 | 0.68 | **0.792** | 89.51 | 93.08 | **86.61** |
| **+ ScaleKV (10%)** | 8.0B | 0.89 | 0.61 | 0.67 | **0.790** | 92.49 | 88.92 | **86.49** |

**Compression Baselines.** We assessed ScaleKV's performance mainly against four representative KV cache compression baselines: Sliding Window Attention [1] (retains local window of most recent tokens), StreamingLLM [62] (keeps attention sinks (initial tokens) and recent tokens), SnapKV [30] (clusters tokens based on attention scores) and PyramidKV [3] (employs a fixed pyramid-shaped allocation strategy across transformer layers).

## 4.2 Main Results

**Comparison with Compression Baselines**. Table 1 presents our evaluation on the MS-COCO 2017 validation set [32], focusing on pixel-level consistency with the original outputs. ScaleKV consistently outperforms all baselines across different memory budgets, with significant improvements in FID, LPIPS, and PSNR metrics.

At the most constrained budget (4%), ScaleKV achieves FID reductions of 31.2% and 48.5% compared to the next best baseline for Infinity-2B and Infinity-8B. The performance gap widens at higher budgets, with ScaleKV achieving FID scores of 1.82 and 1.45 at 20% budget, representing substantial improvements over all competitors. The LPIPS results further validate these findings, with ScaleKV achieving scores of 0.08 and 0.06 at 20% budget for the two models, compared to PyramidKV's 0.11 and 0.10, indicating better perceptual similarity to the original outputs.

Each baseline exhibits specific limitations in the VAR context: Sliding Window Attention and StreamingLLM employ static policies that lose information carried by middle tokens, resulting in poor performance at low budgets. SnapKV's clustering strategy helps preserve some image coherence but cannot effectively prioritize critical tokens across different scales. Notably, PyramidKV's fixed allocation pattern offers limited improvement over SnapKV and sometimes produces worse results

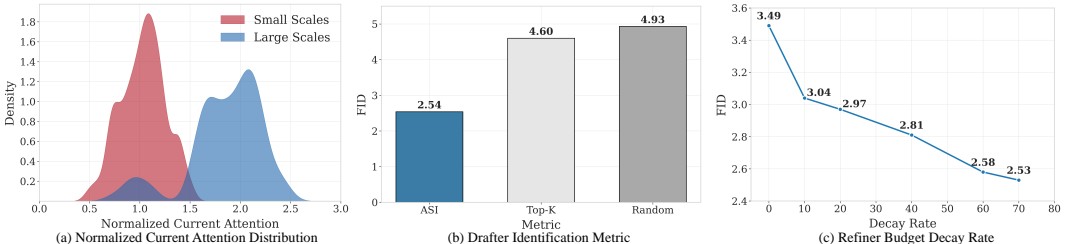

Figure 6: (a) Kernel Density Estimation of normalized current attention scores at small scales $(r_2, r_3, r_4)$ and large scales $(r_{10}, r_{11}, r_{12})$. (b) Ablation experiments on using different drafter identification metrics. (c) Ablation experiments on the impact of refiner budget decay rate.

(e.g., 4% budget on Infinity-2B with FID 5.51 vs. SnapKV's 5.10), confirming that predetermined allocation strategies do not generalize well to VAR's scale-dependent attention behaviors.

**Results on GenEval and DPG**. To further validate the perceptual quality and semantic understanding capabilities of our compressed models, we conducted evaluations on two established benchmarks: GenEval [13] and DPG [20]. Table 2 presents these results, comparing our ScaleKV-compressed models against state-of-the-art image generation models, including diffusion models [43, 5, 2] and autoregressive models [51, 63, 58, 52]. The results demonstrate that ScaleKV preserves semantic understanding remarkably well despite substantial KV cache reduction. For Infinity-2B, our method delivers exceptional performance that matches the full model (83.01 vs. 83.06 on DPG), with the GenEval score even showing a slight improvement from 0.725 to 0.730, while using only 10% of the original KV cache. Similarly, for Infinity-8B, ScaleKV maintains nearly identical performance (0.790 vs. 0.792 on GenEval, and 86.49 vs. 86.61 on DPG). This minimal performance degradation is particularly noteworthy given that Infinity models already outperform most existing approaches on these benchmarks, including larger models like DALL-E 3 and Emu3-8.5B. ScaleKV-compressed Infinity-8B maintains this superior performance while requiring only 8.5 GB of KV cache memory, a dramatic reduction from the original 85 GB.

**Qualitative Results**. We provide an extensive qualitative comparison between the Infinity-8B model with full KV cache and our proposed ScaleKV, with varying budgets of 4%, 10%, and 20%. As illustrated in Figure 5, our approach achieves significant memory optimization, with only minimal quality degradation that is nearly imperceptible to the human eye. Even at a compression rate of 25 times, the generated images maintain exceptionally high quality and accurate semantic information.

### 4.3 Analytical Experiments

**Attention Distributions Across Scales.** We quantify attention pattern variations throughout the multi-scale generation using *normalized current scale attention*, defined as the average attention per token within the current scale to the average attention across the entire sequence, to capture contribution of the cached tokens to generation. Figure 6(a) demonstrates the kernel density estimation (KDE) of normalized current attention, revealing distinct distributions across small scales $(r_2, r_3, r_4)$ and large scales $(r_{10}, r_{11}, r_{12})$. Small scales exhibit an approximately uniform distribution, indicating broad context utilization without strong selectivity. In contrast, large scales concentrate around higher attention values, suggesting focused information selection. This progression reflects an evolution from global information aggregation in early scales to selective attention refinement in later scales.

**Impact of Drafter Identification Metric.** Figure 6(b) demonstrates the comparative effectiveness of different metrics for drafter identification. Our proposed Attention Selectivity Index (ASI) (Equation 2) combines the attention score ratio in the current token map and the Top-K ratio in history sequence. When evaluated on Infinity-2B with a 10% cache budget constraint, ASI achieves an FID score of 2.53, representing a substantial 42.5% improvement over using only the Top-K ratio (4.60). This significant performance gap validates that our comprehensive attention pattern analysis effectively identifies layers requiring larger cache allocation, enabling optimal resource distribution.

**Impact of Refiner Budget Decay Rate.** Figure 6(c) shows the effectiveness of refiner budget decay strategy (Equation 4) under a 10% budget constraint (650 tokens per head/layer). With an initial refiner budget of 600 tokens, we observe a consistent improvement in FID from 3.49 to 2.53 as decay

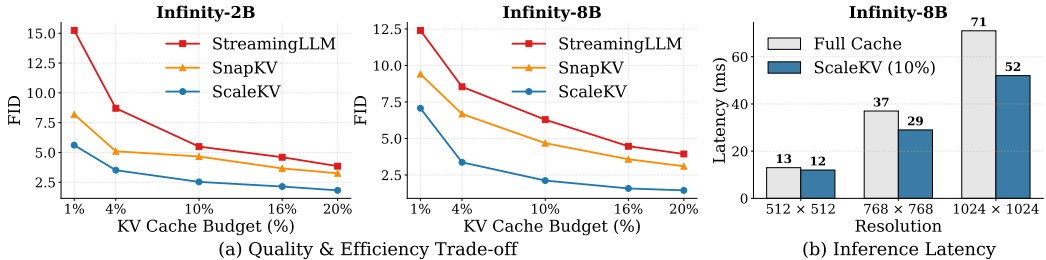

Figure 7: (a) FID under different KV cache budgets. (b) Inference latency for different resolutions.

Table 3: Memory usage comparison across different batch sizes

| Method | Memory Consumption↓ | | | |
|---|---|---|---|---|
| | Running | KV Cache | Params | Total |
| Infinity (bs=1) | 1.1GB | 10.5GB | 19.5GB | 31.4GB |
| +ScaleKV (10%) | 0.8GB | **1.1GB** | 19.5GB | 21.4GB |
| Infinity (bs=8) | - | 85GB | 19.5GB | OOM (>100GB) |
| +ScaleKV (10%) | 21.1GB | **8.5GB** | 19.5GB | 49.2GB |
| Infinity (bs=16) | - | 170GB | 19.5GB | OOM (>100GB) |
| +ScaleKV (10%) | 42.4GB | **17.1GB** | 19.5GB | 78.8GB |

rate increases from 0 to 70, confirming our observation that refiner attention becomes increasingly focused at higher scales, requiring fewer resources. The monotonic improvement also validates our drafter identification method, as reallocating cache capacity from refiners to drafters consistently enhances generation quality, indicating accurate identification of layers with divergent cache demands.

**Quality-Efficiency Trade-off.** We evaluated the quality-efficiency trade-off of ScaleKV against established compression methods across different cache budgets, as shown in Figure 7(a). For both Infinity-2B and Infinity-8B models, ScaleKV consistently achieves the lowest FID at all budget constraints, with performance gap widening at more restrictive memory allocations. These results demonstrate that ScaleKV effectively preserves generation quality even under severe memory constraints, making it adaptable to diverse deployment scenarios with varying computational resources.

**Memory Efficiency and Time Cost Analysis.** In Table 3, we present a comprehensive analysis of memory consumption during the Infinity-8B model inference process. The KV cache of the Infinity model is the largest memory consumer, requiring approximately 10 times the memory needed for the model's decoding operation due to the significantly extended sequence length. Our proposed ScaleKV drastically reduces the KV cache memory requirements, compressing it to 10% of the original model. Moreover, as batch size increases, the memory savings with ScaleKV become even more pronounced. We are able to generate images with a large batch size of 16 using less than 80GB total memory, whereas the KV cache alone of the original model requires 170GB during inference.

While primarily developed to improve memory efficiency, ScaleKV also delivers notable inference acceleration by reducing tensor access and transfer operations. Figure 7(b) illustrates how inference latency increases substantially with image resolution due to exponential growth in the token sequence. Our method achieves up to 1.25× speedup on a single NVIDIA H20 GPU, with performance gains becoming more pronounced as resolution increases. These results demonstrate ScaleKV's potential for deployment in resource-constrained environments and scaling VAR models to ultra-high resolutions such as 4K, which would otherwise be limited by memory bottlenecks and inference latency.

## 5 Conclusion

This work introduces ScaleKV, a novel KV cache compression framework for Visual Autoregressive modeling that effectively addresses the memory bottlenecks in high-resolution image generation. By implementing scale-aware layer budget allocation, ScaleKV enables adaptive cache management tailored to the specific demands of each layer across scales. Through extensive experimentation, our method demonstrates a superior efficiency-quality trade-off, enabling efficient deployment in resource-constrained environments and facilitating the scaling of VAR models to ultra-high resolutions.

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

## Technical Appendices and Supplementary Material

In this document, we provide supplementary material that we could not fit into the main manuscript due to the page limit. It includes detailed explanations, visualization results, and quantitative experiments.

## A    Robustness to Calibration Data

To evaluate the stability of our drafter/refiner identification method across varying calibration set sizes, we conducted an ablation study using prompts sampled from the LAION-Art dataset. We systematically evaluated ScaleKV's performance using calibration sets ranging from a single prompt to 128 prompts, measuring the resulting FID scores on the MS-COCO validation set. As demonstrated in Figure 8, the FID score remains stable at 2.53 with zero standard deviation across the entire range of calibration set sizes. This exceptional consistency indicates that the attention patterns distinguishing drafters from refiners represent fundamental architectural properties of VAR models rather than dataset-specific characteristics. The immediate convergence with even a single calibration sample demonstrates that our Attention Selectivity Index effectively captures the intrinsic scale-dependent attention behaviors—dispersed attention for drafters and concentrated attention for refiners—without requiring extensive statistical sampling. These findings validate that ScaleKV's layer categorization is both theoretically sound and practically robust.

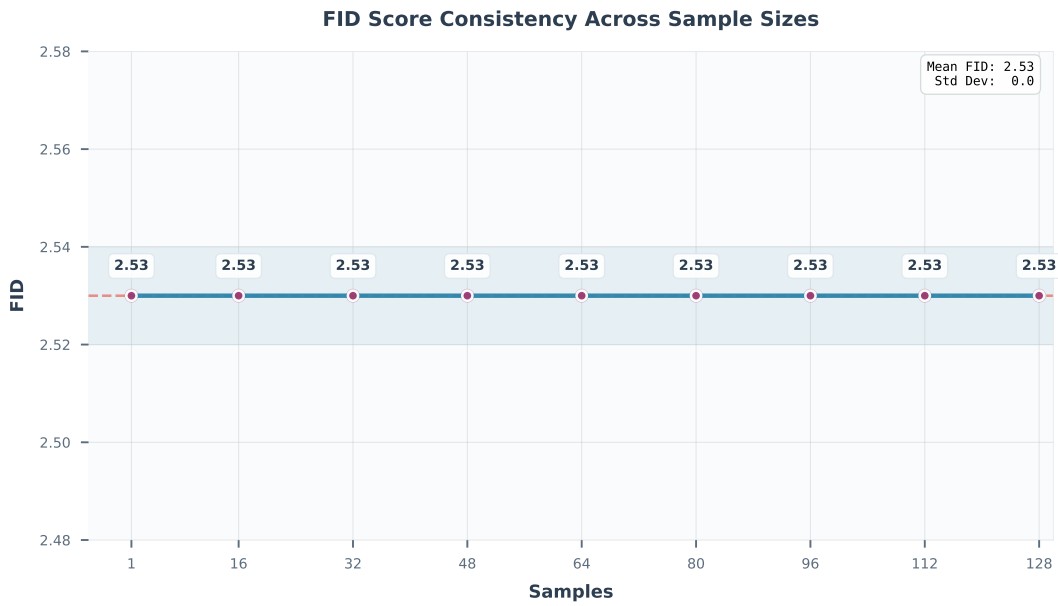

Figure 8: FID score consistency across calibration set sizes. ScaleKV maintains stable performance (FID = 2.53, $\sigma$ = 0.0) from 1 to 128 calibration samples, demonstrating the robustness of our drafter/refiner identification method.

## B    Attention Map Visualization

Figures 9 and 10 present representative attention maps from transformer layers identified as drafters and refiners at scale 7, providing empirical validation for our layer categorization framework. The drafter layer (Block 23) exhibits distinctly dispersed attention patterns across multiple attention heads, with weights distributed broadly across the spatial dimensions to capture global contextual information from preceding scales. This dispersed mechanism enables comprehensive integration of hierarchical features, justifying the larger cache allocation for such layers. In contrast, the refiner layer (Block 29) demonstrates highly concentrated attention patterns, with attention heads focusing predominantly on localized spatial regions within the current token map. These contrasting attention behaviors provide strong empirical evidence for our differentiated cache management strategy: drafters require

substantial cache capacity to maintain broad contextual access while refiners can operate effectively with significantly reduced cache allocation due to their localized processing nature.

## C  Additional Qualitative Results

Figures 11 and 12 present comprehensive galleries of images generated by ScaleKV-compressed Infinity-8B and Infinity-2B models, respectively, demonstrating the practical effectiveness of our compression framework across diverse visual content. These compressed models operate with merely 10% of the original KV cache memory requirement, yet maintain exceptional generation quality across various image categories including natural scenes, objects, portraits, and artistic compositions. These results demonstrate that ScaleKV achieves substantial memory reduction without compromising the generative capabilities of VAR models, making high-quality image synthesis feasible in memory-constrained deployment scenarios.

## D  Limitations

In this analysis, we critically examine the constraints of our methodology.

First, while ScaleKV demonstrates robust compression performance across models of varying capacities, evaluation on larger VAR models would provide additional insights into our method's scalability. Due to the limited availability of large-scale models, our evaluation was restricted to Infinity-8B, currently the largest available VAR model. Testing on models with greater capacity, such as those with 20B parameters, would enable more comprehensive assessment of ScaleKV's scalability. Second, ScaleKV functions as a post-training KV cache compression solution that relies on pre-trained VAR models and mirrors the original model's outputs. Therefore, if the baseline quality of the original VAR models is unsatisfactory, achieving high-quality results with our method could be challenging.

## E  Societal impacts

This work introduces a new KV cache compression framework for VAR models that addresses critical memory bottlenecks in high-resolution image generation. By reducing memory requirements to 10% of the original capacity while maintaining generation quality, our method enhances the accessibility of advanced image synthesis technologies and carries several important societal implications. First, it democratizes access to high-quality image generation by enabling deployment on consumer-grade hardware and edge devices, thereby benefiting creative industries and educational institutions that previously lacked the computational resources for such applications. Second, the reduced memory footprint results in lower energy consumption during inference, contributing to more sustainable AI deployment practices. Third, by enabling ultra-high resolution generation at scales up to 4K, our framework creates new opportunities for professional content creation, medical imaging, and scientific visualization applications.

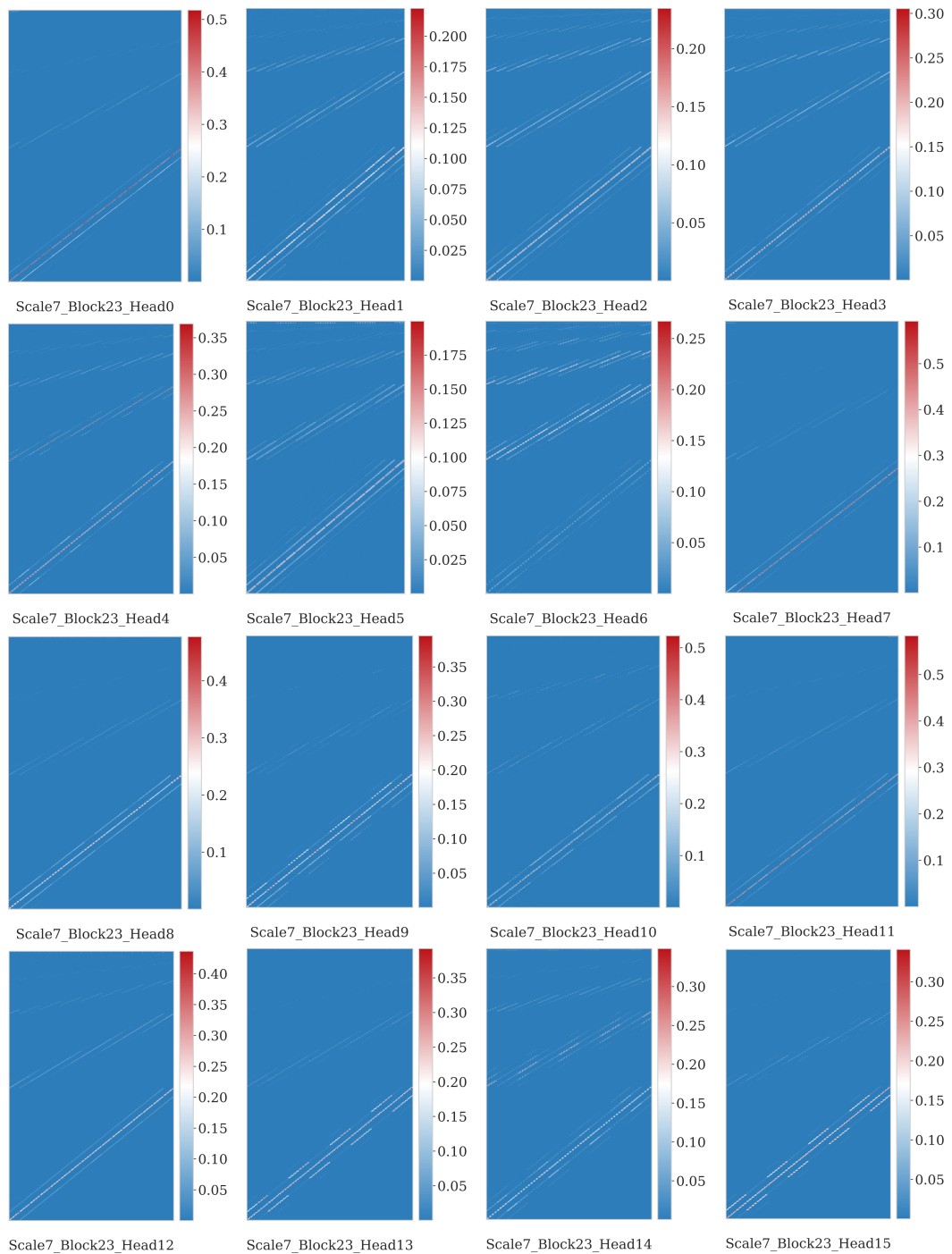

Figure 9: Visualization of Drafter Layer Attention Maps.

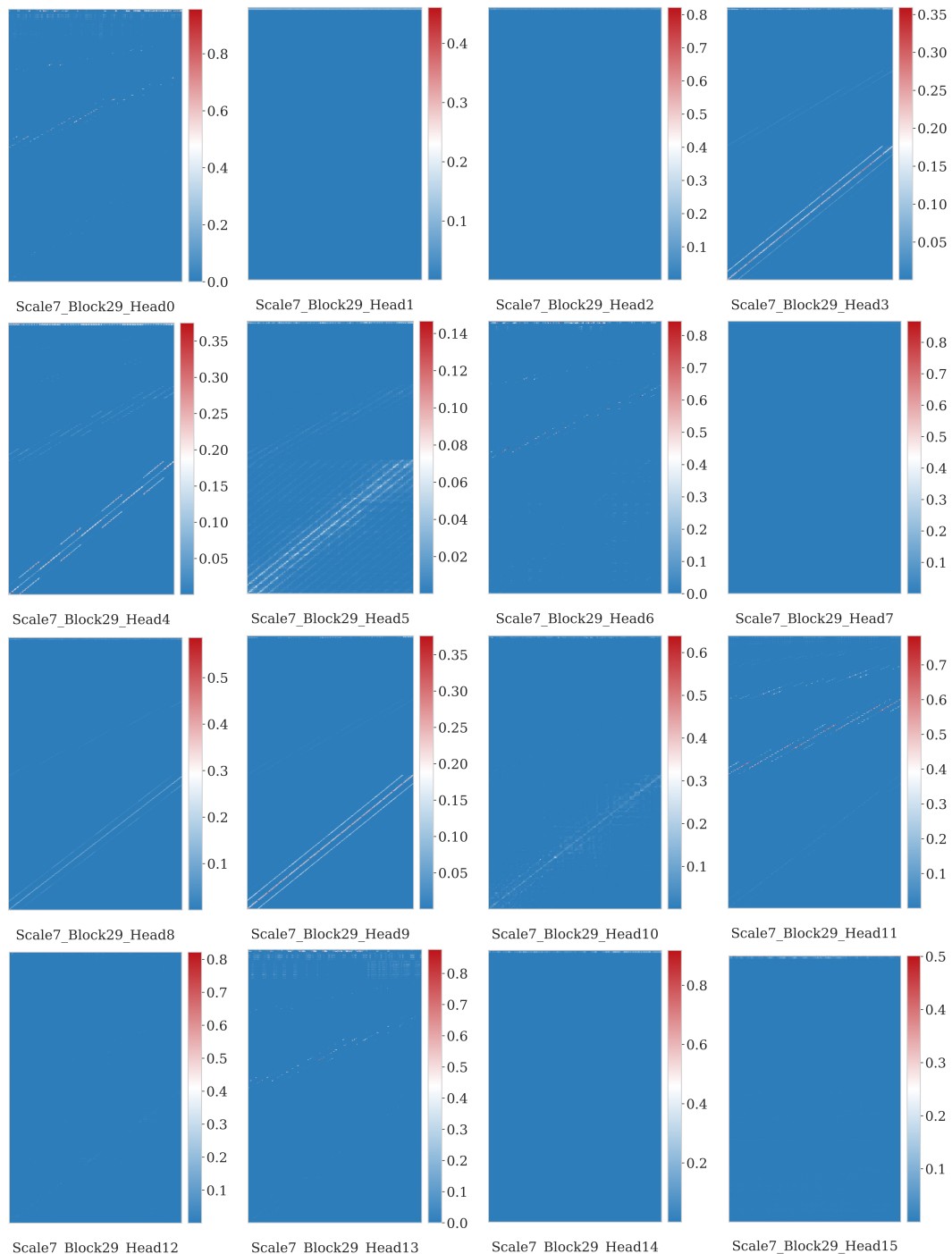

Figure 10: Visualization of Refiner Layer Attention Maps.

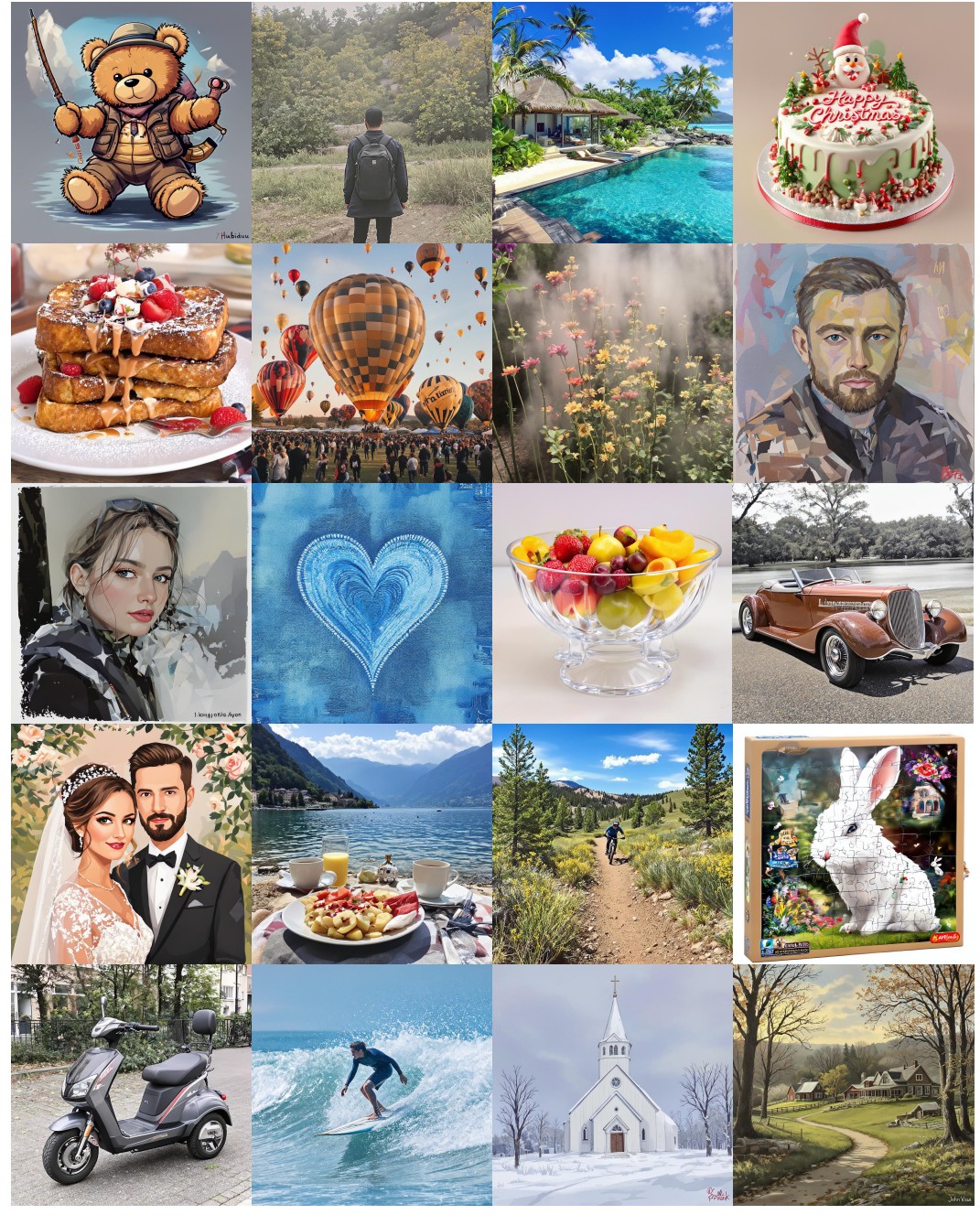

Figure 11: Generated Images from ScaleKV-Compressed Infinity-8B.

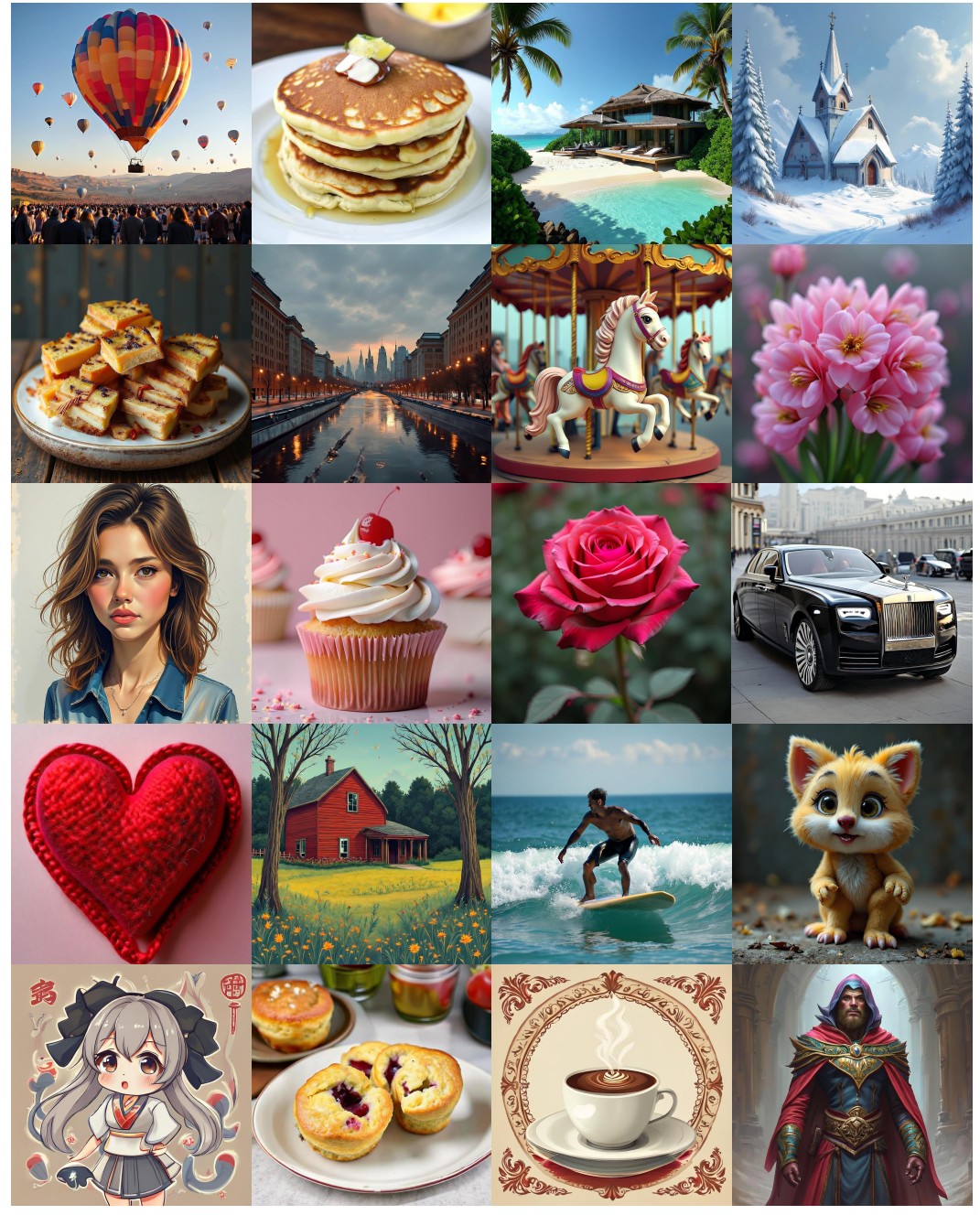

Figure 12: Generated Images from ScaleKV-Compressed Infinity-2B.

