# OpenReview forum: "Memory-Efficient Visual Autoregressive Modeling with Scale-Aware KV Cache Compression"
_NeurIPS.cc/2025/Conference — NeurIPS 2025 poster_

### Official Review · Reviewer_PZnV · 2025-06-23

**Clarity:** 3
**Significance:** 3
**Originality:** 3
**Rating:** 4
**Confidence:** 4

**Summary:**

This paper focuses on improving the inference efficiency for Visual Autoregressive (VAR) modeling, proposing ScaleKV, a novel KV cache compression framework tailored for VAR architecture. Specifically, the proposed ScaleKV categorizes transformer layers into two different groups: drafters and refiners, designing different cache management for each group. Experiments validate its effiectiveness of reducing the required KV cache memory for VAR.

**Questions:**

Except for the weaknesses, the following questions are encouraged to answer:

- Some works share very similar ideas of this paper for improving VAR's generation efficiency, for example [1]. I recommend the authors to discuss with these works, if considering they are concurent works.
- As the rapaid development for AR image models, it is necessary to include more representative AR image models like [2][3].
- Including the original un-conpressed VAR model's performance in Table 1 and Table 2 will help authors to a more comprehensive understanding.
- Add the difference map between generation results of compressed and un-compressed models will be helpful, such as in Figure 1.
- The identification of Drafter and Refiner Layers are decided by the value of Attention Selectivity Index, will there exist a hybrid layer arounding the identification threshold?

[1] Head-Aware KV Cache Compression for Efficient Visual Autoregressive Modeling
[2] Towards Accurate Image Coding: Improved Autoregressive Image Generation With Dynamic Vector Quantization
[3] An Image is Worth 32 Tokens for Reconstruction and Generation

**Ethical Concerns:**

["NO or VERY MINOR ethics concerns only"]

**Final Justification:**

I appreciate the authors’ responses. After reading the authors’ responses and other reviews, I’ve decided to keep my positive score. However, the result on LlamaGen shows that the proposed method exhibit notiable performance drop when generalize the other kinds of AR models (2.5 drop on the overall score). Though I agree the proposed method is valuable for the VAR models, as also pointed by other reviewers that its generalization remain a concern. Therefore I keep my original score, and the generalization limit it from higher scores.

**Limitations:**

See weaknesses and questions.

**Quality:**

3

**Strengths And Weaknesses:**

Strengths:
- Good analysis on the attention pattern of VAR models;
- Strong results on reducing the KV cache memory for VAR models;

Weaknesses:
- The proposed method is specially designed for VAR. Therefore, considering the rapid development of AR image generation, the effectiveness for future AR models maybe limited.
- Too many hyper-parameters making the proposed method much complex and less easy to extend and generalize.

---

> ### Author Rebuttal · Authors · 2025-07-30
>
> We extend our gratitude for your insightful feedback and suggestions, which have helped us improve the clarity and completeness of our work.
>
> > **Q1:** The proposed method is specially designed for VAR. Therefore, considering the rapid development of AR image generation, the effectiveness for future AR models maybe limited.
>
> **A1:** Thank you for the question. Our method generalizes beyond VAR to AR architectures. We demonstrate this by evaluating ScaleKV on LlamaGen [1], an image autoregressive model, showing consistent performance with minimal degradation while achieving significant memory reduction:
>
> | Method | Budget | Global | Entity | Attribute | Relation | Other | Overall ↑ |
> |:---|:---|:---|:---|:---|:---|:---|:---|
> | Full KV | 100% | 77.85 | 76.61 | 76.12 | 79.52 | 78.23 | 65.00 |
> | ScaleKV | 50% | 77.50 | 75.50 | 73.91 | 77.11 | 79.35 | 62.50 |
>
> *Table 1: DPG score of ScaleKV-compressed LlamaGen model*
>
> [1] Autoregressive model beats diffusion: Llama for scalable image generation
>
> > **Q2:** Too many hyper-parameters making the proposed method much complex and less easy to extend and generalize.
>
> **A2:** Thank you for the valuable feedback. Our method has two tunable hyper-parameters: Top-K scores in drafter/refiner classification and observation window size in KV cache selection. Both show robust performance across wide ranges:
>
> | Top-K | FID ↓ | LPIPS ↓ | PSNR ↑ |
> |:---|:---|:---|:---|
> | 1% | 2.58 | 0.11 | 22.79 |
> | 5% | 2.58 | 0.11 | 22.79 |
> | 10% | 2.58 | 0.11 | 22.79 |
> | 20% | 2.59 | 0.11 | 22.77 |
>
> *Table 2: Ablation of Top-K score*
>
> | Window | FID ↓ | LPIPS ↓ | PSNR ↑ |
> |:---|:---|:---|:---|
> | 4 | 2.58 | 0.11 | 22.81 |
> | 8 | 2.61 | 0.11 | 22.78 |
> | 16 | 2.61 | 0.11 | 22.79 |
> | 32 | 2.58 | 0.11 | 22.79 |
>
> *Table 3: Ablation of observation window size*
>
> This stability demonstrates good generalizability and practical usability.
>
> > **Q3:** Some works share very similar ideas of this paper for improving VAR's generation efficiency, for example [1]. I recommend the authors to discuss with these works, if considering they are concurrent works?
>
> **A3:** Thank you for raising this important point. While both methods optimize the KV cache for VAR models, HACK [1] allocates fixed budgets by classifying individual attention heads, whereas ScaleKV dynamically categorizes entire layers as "drafters" or "refiners" to adaptively manage memory at each generation scale. Since HACK hasn't been open-sourced yet, we directly report their DPG score from the original paper.
>
> | Method | Budget | Global | Relation | Overall ↑ |
> |:---|:---|:---|:---|:---|
> | Infinity(8B) | 100% | 89.51 | 93.08 | 86.61 |
> | HACK | 10% | - | - | 85.74 |
> | ScaleKV | 10% | 92.49 | 88.92 | 86.49 |
>
> *Table 4: Comparison of ScaleKV and HACK on DPG benchmark*
>
> [1] Head-Aware KV Cache Compression for Efficient Visual Autoregressive Modeling.
>
> > **Q4:** As the rapid development for AR image models, it is necessary to include more representative AR image models like [2][3].
>
> **A4:** We appreciate this constructive feedback. We have evaluated ScaleKV on LlamaGen in A1 and Table 1, demonstrating effectiveness on large-scale text-to-image AR models beyond VAR framework. This validates our method's broad applicability to the AR family.
>
> > **Q5:** Including the original uncompressed VAR model's performance in Table 1 and Table 2 will help authors to a more comprehensive understanding.
>
> **A5:** We appreciate this observation. We have included performance of the uncompressed models (Infinity-2B and 8B) in the paper's Table 2. Table 1 specifically measures consistency between compressed and original uncompressed models, so only compressed model metrics (FID, LPIPS and PSNR) are relevant there.
>
> > **Q6:** Add the difference map between generation results of compressed and uncompressed models will be helpful, such as in Figure 1.
>
> **A6:** Thank you for this helpful suggestion. We will include difference maps in the revision to provide visual insights into generation quality preservation.
>
> > **Q7:** The identification of Drafter and Refiner Layers are decided by the value of Attention Selectivity Index, will there exist a hybrid layer around the identification threshold?
>
> **A7:** Thank you for the valuable comment, this is indeed a question that needs further explanation. We use a strict threshold ensuring layers are classified as drafters only when cache requirements are definitively large. This conservative approach eliminates ambiguous hybrid cases, ensuring robust layer categorization.

---

### Official Review · Reviewer_KeK9 · 2025-06-25

**Clarity:** 3
**Significance:** 2
**Originality:** 3
**Rating:** 4
**Confidence:** 2

**Summary:**

This paper addresses the KV cache compression problem in Visual Autoregressive (VAR) models, proposing ScaleKV, a framework that categorizes transformer layers into "Drafters" and "Refiners" for scale-aware cache budget allocation. It achieves a 10× reduction of KV cache from 85GB to 8.5GB for the Infinity-8B model with negligible quality degradation (GenEval score remains 0.79, DPG score drops marginally from 86.61 to 86.49). The technical approach is clear and experimentally validated, but its impact is limited by the niche nature of VAR models in the visual generation landscape.

**Questions:**

Please refer to the weakness part

**Ethical Concerns:**

["NO or VERY MINOR ethics concerns only"]

**Final Justification:**

Thanks for the authors' response. I am happy to Borderline accept this paper.

**Limitations:**

yes

**Quality:**

3

**Strengths And Weaknesses:**

Strengths:
1. Technical Insights：The framework leverages two critical insights: varying cache demands across transformer layers and scale-dependent attention patterns in VAR.
2. The paper is well writen and the authors conduct rigorous experiment to verify the effectiveness.


Weakness
1. Niche Application Scope: VAR models (e.g., Infinity) are not the mainstream in text-to-image generation, which is currently dominated by diffusion models (e.g., SDXL, SD3, SANA, ...). The paper does not demonstrate whether ScaleKV’s core strategies (e.g., layer classification, scale-aware allocation) can generalize to non-VAR frameworks, limiting practical impact.

---

> ### Author Rebuttal · Authors · 2025-07-30
>
> We're grateful for the feedback you've provided.
>
> > **Q1:** Niche Application Scope: VAR models (e.g., Infinity) are not the mainstream in text-to-image generation, which is currently dominated by diffusion models (e.g., SDXL, SD3, SANA, ...). The paper does not demonstrate whether ScaleKV’s core strategies (e.g., layer classification, scale-aware allocation) can generalize to non-VAR frameworks, limiting practical impact
>
> **A1:** Thank you for the valuable feedback. We acknowledge that diffusion models currently dominate text-to-image generation, but we believe ScaleKV addresses fundamental challenges that extend beyond VAR models:
>
> **Broader Applicability Beyond VAR:** Our method generalizes beyond VAR to traditional AR frameworks. We validate this by applying ScaleKV to LlamaGen [1], an image autoregressive model, demonstrating robust performance with minimal quality loss while achieving substantial memory savings:
>
> | Method | Budget | Global | Entity | Attribute | Relation | Other | Overall ↑ |
> |:---|:---|:---|:---|:---|:---|:---|:---|
> | Full KV | 100% | 77.85 | 76.61 | 76.12 | 79.52 | 78.23 | 65.00 |
> | ScaleKV | 50% | 77.50 | 75.50 | 73.91 | 77.11 | 79.35 | 62.50 |
>
> *Table 1: DPG score of ScaleKV-compressed LlamaGen model*
>
> **Strategic Focus on VAR:** VAR models are gaining remarkable attention due to their efficiency, scalability, and potential for unified vision understanding and generation. ScaleKV effectively solves the memory bottleneck that makes high-resolution modeling challenging, enabling these promising architectures to scale to ultra-high resolutions such as 4K.
>
> **Future Extensibility:** Our drafter/refiner classification and scale-aware allocation principles provide a methodology that could adapt to future architectures, including potential hybrid approaches combining visual autoregressive and diffusion techniques.
>
> [1] Autoregressive model beats diffusion: Llama for scalable image generation.

---

### Official Review · Reviewer_KGmY · 2025-06-26

**Clarity:** 3
**Significance:** 3
**Originality:** 3
**Rating:** 5
**Confidence:** 4

**Summary:**

The paper highlights that in Visual Autoregressive (VAR) models, the KV cache experiences exponential growth during inference due to the multi-scale generation approach. To address this problem, the paper introduces a KV cache allocation method called ScaleKV, which classifies Transformer layers into "Drafters" and "Refiners". Experimental results on the Infinity-8B and 2B models show that this KV cache compression effectively resolves memory pressure while maintaining accuracy.

**Questions:**

1. Why not perform the drafter/refiner identification on a per-sample basis? I am concerned about the method's adaptability to other models. Also, when the calibration set size varies, do other metrics besides FID also remain consistent?
2. What is the dominant component of the Attention Selectivity Index (ASI)—the 'current' attention ratio or the 'history' Top-K ratio, and how are they balanced? How is the Top-K' parameter selected, and what is the basis for choosing the observation window size (mentioned as typically 16 tokens)?
3. The baselines for comparison might not be recent enough; could you compare against more current cross-layer KV allocation methods? I'm also wondering whether a more granular, per-layer management strategy would be better than the current simple distinction between 'drafters' and 'refiners'.
4. I would like to know the specific details of the inference latency measurements in Figure 7b. Can ScaleKV be seamlessly integrated with standard optimizations like Flash Attention? Additionally, what was the image resolution used for the main results presented in the tables?

**Ethical Concerns:**

["NO or VERY MINOR ethics concerns only"]

**Final Justification:**

I appreciate the authors' solid rebuttal, which has addressed most of my concerns. I have also read other reviews and the authors' responses to them. I agree with other reviewers on accepting the paper. Therefore, my final rating is an Accept. Congrats!

**Limitations:**

yes

**Quality:**

4

**Strengths And Weaknesses:**

### Strengths

* The paper focused on a meaningful problem of reducing the exponential KV cache growth of a popular generation paradigm, Visual Autoregressive (VAR) models. I like the problem and the scope.
* The proposed Drafter/Refiner is interesting and novel.
* The experimental results are promising and improvements are significant (if reproducible). I appreciate the authors' attachment of code in the supplementary materials.

### Weaknesses

* How can this method generalize to other datasets? Currently, the method's static, binary classification of layers has only been evaluated on the Infinity model family and requires further validation for its generalization across diverse inputs and other model architectures. I would like to see some discussions or empirical results.
* It is still unclear about selecting core hyperparameters, such as the K' in the ASI formula and the observation window size. Some key experimental setup details should be complete.
* The paper did not discuss the method's integration and compatibility with critical performance optimizations, such as Flash Attention.

---

> ### Author Rebuttal · Authors · 2025-07-30
>
> Thank you very much for your review of our manuscript and attachment. We appreciate your time and effort in evaluating our work.
>
> > **Q1:** How can this method generalize to other datasets? Currently, the method's static, binary classification of layers has only been evaluated on the Infinity model family and requires further validation for its generalization across diverse inputs and other model architectures. I would like to see some discussions or empirical results.
>
> **A1:** Thank you for this valuable feedback. We evaluated ASI robustness across three distinct visual domains using 128 calibration prompts each from HuggingFace datasets: flickr30k_captions_simCSE (human faces), architecture_house_building_prompts_SDXL (architecture details) and nature-dataset (natural landscapes). Results show consistent performance across domains, confirming that drafter-refiner distinctions reflect model characteristics rather than domain-specific patterns. We also demonstrate applicability to traditional AR architectures like LlamaGen (Table 5).
>
> | Dataset | FID ↓ | LPIPS ↓ | PSNR ↑ |
> |:---|:---|:---|:---|
> | Human Face | 2.58 | 0.11 | 22.77 |
> | Architecture | 2.57 | 0.11 | 22.81 |
> | Landscape | 2.57 | 0.11 | 22.79 |
>
> *Table 1: Cross-domain validation results*
>
> > **Q2:** It is still unclear about selecting core hyperparameters, such as the K' in the ASI formula and the observation window size. Some key experimental setup details should be complete.
>
> **A2:** We appreciate your suggestions on clarifying these details. Our method demonstrates robust performance across varying hyperparameters, requiring minimal tuning. Tables 2-3 show stable results across different Top-K values (1%-20%) and window sizes (4-32 tokens).
>
> | Top-K | FID ↓ | LPIPS ↓ | PSNR ↑ |
> |:---|:---|:---|:---|
> | 1% | 2.58 | 0.11 | 22.79 |
> | 5% | 2.58 | 0.11 | 22.79 |
> | 10% | 2.58 | 0.11 | 22.79 |
> | 20% | 2.59 | 0.11 | 22.77 |
>
> *Table 2: Top-K parameter ablation*
>
> | Window | FID ↓ | LPIPS ↓ | PSNR ↑ |
> |:---|:---|:---|:---|
> | 4 | 2.58 | 0.11 | 22.81 |
> | 8 | 2.61 | 0.11 | 22.78 |
> | 16 | 2.61 | 0.11 | 22.79 |
> | 32 | 2.58 | 0.11 | 22.79 |
>
> *Table 3: Observation window size ablation*
>
> > **Q3:** The paper did not discuss the method's integration and compatibility with critical performance optimizations, such as Flash Attention.
>
> **A3:** Thank you for pointing this out. ScaleKV is fully compatible with FlashAttention. We report the inference latency of Infinity-8B measured on a single NVIDIA A100-SXM4-80GB GPU. Table 4 shows combined optimization achieves 2.7× speedup while maintaining compression benefits.
>
> | Method | Budget | Latency (ms) |
> |:---|:---|:---|
> | Full KV | 100% | 77 |
> | ScaleKV | 10% | 46 |
> | ScaleKV + FlashAttention | 10% | 29 |
>
> *Table 4: Inference latency comparison (Infinity-8B)*
>
> > **Q4:** Why not perform the drafter/refiner identification on a per-sample basis? I am concerned about the method's adaptability to other models.
>
> **A4:** Thanks for raising this important concern. Our drafter/refiner identification process is conducted before model deployment. Performing calibration on a per-sample basis would require capturing full attention maps for each sample, which would eliminate the computational benefits of KV cache compression.
>
> Our compression method is also applicable to traditional AR image generation frameworks, including LlamaGen [1].
>
> | Method | Budget | Global | Entity | Attribute | Relation | Other | Overall ↑ |
> |:---|:---|:---|:---|:---|:---|:---|:---|
> | Full KV | 100% | 77.85 | 76.61 | 76.12 | 79.52 | 78.23 | 65.00 |
> | ScaleKV | 50% | 77.50 | 75.50 | 73.91 | 77.11 | 79.35 | 62.50 |
>
> *Table 5: DPG scores of ScaleKV-compressed LlamaGen model*
>
> [1] Autoregressive model beats diffusion: Llama for scalable image generation.
>
> > **Q5:** Also, when the calibration set size varies, do other metrics besides FID also remain consistent?
>
> **A5:** Thank you for this question. We tested calibration sizes from 1 to 128 prompts using GPT-4o generated data. Table 6 demonstrates stable performance regardless of calibration size, confirming that minimal data suffices due to fundamental attention pattern differences and cache requirements between layer types.
>
> | Size | FID ↓ | LPIPS ↓ | PSNR ↑ |
> |:---|:---|:---|:---|
> | 1 | 2.58 | 0.11 | 22.78 |
> | 4 | 2.57 | 0.11 | 22.79 |
> | 16 | 2.57 | 0.11 | 22.81 |
> | 64 | 2.57 | 0.11 | 22.81 |
> | 128 | 2.57 | 0.11 | 22.81 |
>
> *Table 6: Calibration data size ablation*
>
> > **Q6:** What is the dominant component of the Attention Selectivity Index (ASI)—the 'current' attention ratio or the 'history' Top-K ratio, and how are they balanced? How is the Top-K' parameter selected, and what is the basis for choosing the observation window size (mentioned as typically 16 tokens)?
>
> **A6:** Thanks for the valuable comment. The Attention Selectivity Index (ASI) multiplies the current attention ratio and history Top-K ratio, ensuring both components contribute equally to layer classification. The robustness experiments above demonstrate minimal parameter tuning requirements (see A2 and Tables 2-3). We chose 16 tokens as the default window size to ensure fair comparison with existing baselines, as this is the standard setting used in most KV cache compression methods.
>
> > **Q7:** The baselines for comparison might not be recent enough; could you compare against more current cross-layer KV allocation methods?
>
> **A7:** Thanks for the suggestion. We compared ScaleKV against two recent cross-layer KV cache allocation methods: PyramidKV [1] (employs pyramid-shaped budget allocation) and SimLayerKV [2] (attention-based selective cache allocation). ScaleKV significantly outperforms both methods across all metrics.
>
> | Method | FID ↓ | LPIPS ↓ | PSNR ↑ |
> |:---|:---|:---|:---|
> | PyramidKV | 4.52 | 0.16 | 20.92 |
> | SimLayerKV | 3.55 | 0.13 | 22.39 |
> | ScaleKV | 2.57 | 0.11 | 22.81 |
>
> *Table 7: Comparison with cross-layer KV cache allocation baselines*
>
> [1] PyramidKV: Dynamic KV Cache Compression based on Pyramidal Information Funneling
>
> [2] SimLayerKV: A Simple Framework for Layer-Level KV Cache Reduction
>
> > **Q8:** I'm also wondering whether a more granular, per-layer management strategy would be better than the current simple distinction between 'drafters' and 'refiners'.
>
> **A8:** We appreciate this constructive feedback. We implemented a per-layer granular management strategy that allocates individual budgets based on the proposed Attention Selectivity Index (ASI). As shown in Table 8, granular allocation does not provide additional benefits over our current method, supporting our hypothesis that cache demands are similar within groups but different between drafter/refiner groups.
>
> | Method | FID ↓ | LPIPS ↓ | PSNR ↑ |
> |:---|:---|:---|:---|
> | Granular | 2.70 | 0.12 | 22.51 |
> | ScaleKV | 2.57 | 0.11 | 22.81 |
>
> *Table 8: Comparison with per-layer management strategy*
>
> > **Q9:** Additionally, what was the image resolution used for the main results presented in the tables?
>
> **A9:** Thanks for the question. All experiments use 1024×1024 resolution to validate ScaleKV's effectiveness for high-resolution generation, where memory bottlenecks are most critical.

---

> ### Comment · Reviewer_KGmY · 2025-08-02
>
> I appreciate the authors' solid rebuttal, which has addressed most of my concerns. I have also read other reviews and the authors' responses to them. I agree with other reviewers on accepting the paper. Therefore, my final rating is an Accept. Congrats!

---

> > ### Author Response · Authors · 2025-08-02
> >
> > Thanks so much for the encouraging feedback! We will keep refining the draft with all the additional experiments.
> >
> > Best regards, Authors

---

### Official Review · Reviewer_VBDw · 2025-07-02

**Clarity:** 3
**Significance:** 2
**Originality:** 2
**Rating:** 4
**Confidence:** 2

**Summary:**

The paper introduces ScaleKV, a novel KV cache compression framework for Visual Autoregressive (VAR) models, addressing memory bottlenecks arising from exponential KV cache growth during inference. Based on the attention patterns, the proposed method categorizes transformer layers into two functional groups (Drafters and Refiners). Drafters exhibit dispersed attention across multiple scales, requiring substantial cache storage, whereas Refiners have localized attention, requiring minimal storage. The framework implements an adaptive, scale-aware budget allocation strategy, reducing KV cache memory usage to approximately 10% of the original size while maintaining image quality. Experimental validation on Infinity-2B and Infinity-8B VAR models using standard benchmarks (MS-COCO, GenEval, DPG) shows ScaleKV significantly outperforms other state-of-the-art cache compression methods.

**Questions:**

## Questions
1. Architecture variations: The paper's evaluation focuses exclusively on Infinity models. How does ScaleKV perform on VAR models with architectural variations? Specifically, I'm curious about models using cross-attention mechanisms between different scales rather than just self-attention within scales, as these might exhibit fundamentally different attention patterns.
2. Integration with other optimizations: Can ScaleKV be effectively combined with quantization methods (e.g., LiteVAR [1])? Cache compression and reduced precision can either help each other or get in each other’s way.

## Minor Issues:
1. Citation [51] [52] repeats.
2. The number of decimal places in Table 2 are not uniform. Some numbers have 2 decimal places and some numbers have 3 decimal places.
3. Typo in a reference title. There is a missing space in ‘recipefor’ in the title of [22].
4. In Eq. (2), the variable name ASI should use mathrm or text to distinguish with the math symbols.
5. \citet and \citep are not properly used. For example, in Line 163, it should not directly refer to the reference [30]. There should be a noun before it.
6. It is better to have more explanation in the caption of Figure 2 and Figure 3. Otherwise, the information the figures are intended to convey are unclear.

[1] Xie, Rui, et al. "LiteVAR: Compressing Visual Autoregressive Modelling with Efficient Attention and Quantization." arXiv preprint 2024.

**Ethical Concerns:**

["NO or VERY MINOR ethics concerns only"]

**Final Justification:**

The authors have fully addressed my concerns, and I believe my concerns have been adequately addressed. I am satisfied with the answers and will maintain my positive score to reflect my support for acceptance.

**Limitations:**

yes

**Quality:**

3

**Strengths And Weaknesses:**

## Strengths:

1. The visualization is clear and understandable. It is easy for readers to grasp the core idea without diving into formulas. The progressive visualization from Scale 4 to Scale 5 demonstrates how refiners become increasingly concentrated at higher scales, supporting the core hypothesis.
2. The experiments are solid and comprehensive. The paper evaluates the proposed method on multiple popular benchmarks, combining quantitative metrics with side-by-side images, which lends credibility to the claims. Two model scales (Infinity-2B and 8B) are both put through full ablations, comparing against static quotas, random dropping, and learned baselines, which evidences that the authors did extensive groundwork rather than cherry-pick results.
3. It achieves 10x reduction in KV cache memory requirements. Cutting the 85 GB cache of an 8-B model down to 8.5 GB turns “needs multi-GPU offload” into “fits on one high-end card”. Besides raw memory, smaller caches can reduce PCIe traffic and hidden latency spikes, so throughput may rise in practice.


## Weaknesses:
1. Lack of discussion of generalization of the proposed method: While the paper thoroughly evaluates on Infinity models, it provides insufficient discussion of transferability. The evaluation does not include VAR variants with different attention mechanisms. It remains unclear whether the drafter/refiner would hold for architectures with different layer configurations or attention head designs. Additionally, the paper neither analyzes if the ASI metric (Eq. 2) requires adjustment for models with varying attention mechanisms nor discusses its adaptability to future VAR architectures with novel designs.
2. Evaluation on larger models: As acknowledged in Appendix D, the evaluation covers models up to 8B parameters, with Infinity-8B as the largest tested. Thus, empirical validation is missing for truly large-scale (e.g., 20B) models, where attention patterns and drafter/refiner ratios might differ. Additionally, the linear scaling assumption for memory savings remains unverified at larger scales.
3. Trade-off analysis: While Figure 7 (a) provides a trade-off curve showing FID scores across five compression ratios (1%, 4%, 10%, 16%, 20%), the analysis could be more comprehensive. The paper focuses primarily on three ratios (4%, 10%, 20%) in most experiments and does not provide guidance on choosing appropriate compression levels for specific use cases. There is no principled method offered for practitioners to determine optimal compression ratios given their quality constraints. Besides, the paper lacks analysis of which image types or generation tasks are most sensitive to compression. For example, whether architectural details, human faces, or natural landscapes show different degradation patterns.

---

> ### Author Rebuttal · Authors · 2025-07-30
>
> Thank you for your valuable feedback on our work. Your constructive comments on our work are invaluable.
>
> > **Q1:** Lack of discussion of generalization of the proposed method: While the paper thoroughly evaluates on Infinity models, it provides insufficient discussion of transferability. The evaluation does not include VAR variants with different attention mechanisms. It remains unclear whether the drafter/refiner would hold for architectures with different layer configurations or attention head designs.
>
> **A1:** Thank you for this valuable feedback. Our method generalizes beyond VAR architectures. We validate this by applying ScaleKV to LlamaGen [1], an autoregressive image model with fundamentally different attention mechanisms. Table 1 shows consistent performance with minimal degradation while achieving significant memory reduction. We also tested across different model scales (Infinity-2B/8B), confirming robustness across layer configurations in the paper.
>
> | Method | Budget | Global | Entity | Attribute | Relation | Other | Overall ↑ |
> |:---|:---|:---|:---|:---|:---|:---|:---|
> | Full KV | 100% | 77.85 | 76.61 | 76.12 | 79.52 | 78.23 | 65.00 |
> | ScaleKV | 50% | 77.50 | 75.50 | 73.91 | 77.11 | 79.35 | 62.50 |
>
> *Table 1: DPG score of ScaleKV-compressed LlamaGen model*
>
> [1] Autoregressive model beats diffusion: Llama for scalable image generation
>
> > **Q2:** Additionally, the paper neither analyzes if the ASI metric (Eq. 2) requires adjustment for models with varying attention mechanisms nor discusses its adaptability to future VAR architectures with novel designs.
>
> **A2:** We appreciate this important concern. For future VAR architectures that use Grouped-Query Attention (GQA) for efficient computation, ASI should be adjusted to measure attention diversity between entire query groups rather than between individual heads. This involves shifting the core unit of analysis from the head to the group level, as heads within a group are no longer independent.
>
> > **Q3:** Evaluation on larger models: As acknowledged in Appendix D, the evaluation covers models up to 8B parameters, with Infinity-8B as the largest tested. Thus, empirical validation is missing for truly large-scale (e.g., 20B) models, where attention patterns and drafter/refiner ratios might differ. Additionally, the linear scaling assumption for memory savings remains unverified at larger scales.
>
> **A3:** Thanks for the valuable comment. We tested our method on Infinity-8B, currently the largest available VAR model. Results demonstrate improved scaling behavior—performance increases from Infinity-2B to 8B (FID: 2.53→2.12 at 10% compression), suggesting larger models benefit more from ScaleKV due to increased attention redundancy. The linear scaling assumption holds because memory reduction is directly proportional to pruned KV cache tokens. We will evaluate on larger models when available.
>
> > **Q4:** Trade-off analysis: While Figure 7 (a) provides a trade-off curve showing FID scores across five compression ratios (1%, 4%, 10%, 16%, 20%), the analysis could be more comprehensive. The paper focuses primarily on three ratios (4%, 10%, 20%) in most experiments and does not provide guidance on choosing appropriate compression levels for specific use cases. There is no principled method offered for practitioners to determine optimal compression ratios given their quality constraints.
>
> **A4:** Thank you for your constructive suggestion. We provide comprehensive trade-off analysis with clear guidance for practitioners. Table 2 demonstrates ScaleKV's effectiveness across five compression ratios on Infinity-2B, showing a clear quality-efficiency trade-off. Extreme compression levels (1-4%) achieve dramatic memory reduction suitable for resource-constrained environments like mobile devices, while moderate compression (10-16%) maintains high visual quality for standard deployments. Conservative compression (20%) preserves exceptional quality for professional applications requiring high fidelity.
>
> | Budget | KV Cache | FID ↓ | Use Case |
> |:---|:---|:---|:---|
> | 1% | 390MB | 5.91 | Mobile/embedded devices |
> | 4% | 1590MB | 3.51 | Edge computing |
> | 10% | 3900MB | 2.53 | Desktop GPUs |
> | 16% | 6200MB | 2.14 | Workstations |
> | 20% | 7800MB | 1.82 | Cloud deployment |
>
> *Table 2: Quality & efficiency trade-off (Infinity-2B)*
>
> > **Q5:** Besides, the paper lacks analysis of which image types or generation tasks are most sensitive to compression. For example, whether architectural details, human faces, or natural landscapes show different degradation patterns.
>
> **A5:** Thank you for raising this important point. We evaluated ScaleKV across three distinct visual domains using 5,000 prompts each from HuggingFace datasets: flickr30k_captions_simCSE (human faces), architecture_house_building_prompts_SDXL (architectural details) and nature-dataset (natural landscapes). Table 3-5 demonstrate consistent performance across all domains, confirming ScaleKV's robustness for diverse visual content.
>
> | Method | Budget | FID ↓ | LPIPS ↓ | PSNR ↑ |
> |:---|:---|:---|:---|:---|
> | Full KV | 100% | - | - | - |
> | ScaleKV | 10% | 2.02 | 0.08 | 27.41 |
>
> *Table 3: Output consistency on human faces*
>
> | Method | Budget | FID ↓ | LPIPS ↓ | PSNR ↑ |
> |:---|:---|:---|:---|:---|
> | Full KV | 100% | - | - | - |
> | ScaleKV | 10% | 2.34 | 0.18 | 19.80 |
>
> *Table 4: Output consistency on architecture details*
>
> | Method | Budget | FID ↓ | LPIPS ↓ | PSNR ↑ |
> |:---|:---|:---|:---|:---|
> | Full KV | 100% | - | - | - |
> | ScaleKV | 10% | 2.72 | 0.15 | 22.52 |
>
> *Table 5: Output consistency on nature landscapes*
>
> > **Q6:** Architecture variations: The paper's evaluation focuses exclusively on Infinity models. How does ScaleKV perform on VAR models with architectural variations? Specifically, I'm curious about models using cross-attention mechanisms between different scales rather than just self-attention within scales, as these might exhibit fundamentally different attention patterns.
>
> **A6:** Thank you for the question. VAR models employ complex attention mechanisms beyond simple self-attention: causal cross-attention between current scale tokens and previous prefix tokens in KV cache, combined with bidirectional attention within current scale tokens. Our successful evaluation on both VAR and LlamaGen with different mechanisms (discussed in A1 and Table 1) demonstrates broad applicability across diverse attention patterns.
>
> > **Q7:** Integration with other optimizations: Can ScaleKV be effectively combined with quantization methods (e.g., LiteVAR [1])? Cache compression and reduced precision can either help each other or get in each other’s way.
>
> **A7:** Thank you for the valuable suggestion. ScaleKV is designed to be complementary to other optimization techniques. Both ScaleKV and LiteVAR can optimize inference efficiency with negligible quality degradation, so combining them could further enhance overall efficiency. However, since LiteVAR has not been open-sourced yet, we could not test this integration experimentally.
>
> Beyond quantization, ScaleKV could be integrated effectively with FlashAttention [1] to further optimize computational efficiency. We report inference latency of Infinity-8B measured on a single NVIDIA A100-SXM4-80GB GPU in Table 6, showing that the combined approach achieves 2.66× speedup while maintaining the same memory compression benefits.
>
> | Method | Budget | Latency (ms) |
> |:---|:---|:---|
> | Full KV | 100% | 77 |
> | ScaleKV | 10% | 46 |
> | ScaleKV + FlashAttention | 10% | 29 |
>
> *Table 6: Inference latency of Infinity-8B*
>
> [1] FlashAttention: Fast and Memory-Efficient Exact Attention with IO-Awareness
>
> > **Q8:** Minor Issues
>
> **A8:** Thank you for the careful review. We will address all minor issues including typos, formatting inconsistencies, and notation clarifications in the revised manuscript.

---

> > ### Comment · Reviewer_VBDw · 2025-08-04
> >
> > I appreciate the authors’ responses. After reading the authors’ responses and other reviews, I’ve decided to keep my positive score.

---

> > > ### Author Response · Authors · 2025-08-05
> > >
> > > We would like to express our sincere gratitude for the insightful comments! We will improve the quality of our draft following the above suggestions.

---

### Official Review · Reviewer_74F6 · 2025-07-03

**Clarity:** 3
**Significance:** 3
**Originality:** 2
**Rating:** 5
**Confidence:** 4

**Summary:**

The paper introduces ScaleKV, a novel KV cache compression framework for Visual Autoregressive (VAR) modeling, addressing the exponential memory growth issue in multi-scale inference. Key insights reveal varying cache demands across transformer layers and distinct attention patterns at different scales, leading to the classification of layers into "Drafters" (requiring large cache for global context) and "Refiners" (needing minimal cache for local detail processing).

ScaleKV implements scale-aware layer budget allocation, using the Attention Selectivity Index (ASI) to distinguish layer types. Refiner budgets decrease linearly with scale, reallocating saved memory to Drafters. A token selection strategy preserves critical KV states based on attention importance, ensuring spatial coverage with minimal overhead.

Experiments on Infinity-2B/8B models show ScaleKV reduces KV cache memory to 10% (e.g., 85GB to 8.5GB for Infinity-8B) with negligible quality loss (GenEval score remains 0.79, DPG drops marginally from 86.61 to 86.49). It outperforms baselines in FID, LPIPS, and PSNR across memory budgets, achieving up to 1.25× inference speedup.

**Questions:**

refer to weakness

**Ethical Concerns:**

["NO or VERY MINOR ethics concerns only"]

**Final Justification:**

The author has basically solved my problem, and I will upgrade my rating to "accept".

**Paper Formatting Concerns:**

For me, this paper is very well-written and there are no obvious formatting problems.

**Quality:**

3

**Strengths And Weaknesses:**

Strengths:

1/ After carefully reading the paper, I find the proposed ScaleKV method an interesting training-free acceleration work, and the paper is clearly written. The paper identifies several obvious issues in VAR, among which the token sequence and KV cache become heavy at high resolutions and subsequent scales, which is an urgent problem to solve. I have seen many related works optimizing this problem, indicating the importance of this direction. Based on this, the work discovers that different scales exhibit distinct attention patterns. Through this interesting phenomenon, the authors propose a scale-aware KV cache to optimize FLOPs. The overall motivation and experiments are reasonable.

2/ The experimental part is also straightforward. The authors conduct sufficient experiments on GenEval and DPG benchmarks based on the VAR T2I works Infinity 2B & 8B. The experimental results are sufficient and detailed. The authors also compare with classic methods such as Sliding Window, Streaming LLM, and PyramidKV. From the indicators, this optimization basically causes no performance drop.

Weakness:

1/ The Attention Selectivity Index (ASI) requires minimal calibration data (10 prompts) to distinguish Drafters and Refiners. However, the method’s sensitivity to calibration prompt diversity and content is not fully explored. For example, if prompts are biased toward specific visual domains, the layer classification might not generalize to broader scenarios, potentially affecting cache allocation accuracy. Additionally, I am curious whether the experimental conclusions are consistent with different amounts of calibration data. ScaleKV only tested on 10 prompts to determine the drafters and refiners—are there more conclusions supporting this?

2/ Regarding actual running speed rather than computational load or GPU usage, I noticed in Fig7(b) that the latency for 1024×1024 resolution is only 52ms vs. 71ms compared to full cache. Are there any optimization potentials or observations here? What other drawbacks does the current KV cache method have?

---

> ### Author Rebuttal · Authors · 2025-07-30
>
> We sincerely appreciate the valuable feedback and constructive suggestions. Thanks so much for taking time and effort to review our paper.
>
> > **Q1:** The Attention Selectivity Index (ASI) requires minimal calibration data (10 prompts) to distinguish Drafters and Refiners. However, the method's sensitivity to calibration prompt diversity and content is not fully explored. For example, if prompts are biased toward specific visual domains, the layer classification might not generalize to broader scenarios, potentially affecting cache allocation accuracy.
>
> **A1:** Thanks for the valuable feedback. We evaluated ASI robustness across three distinct visual domains using 128 calibration prompts each from HuggingFace datasets: flickr30k_captions_simCSE (human faces), architecture_house_building_prompts_SDXL (architecture details) and nature-dataset (natural landscapes). Table 1 shows consistent performance across all domains, confirming that drafter-refiner distinctions reflect intrinsic VAR model characteristics rather than domain-specific patterns.
>
> | Dataset | FID ↓ | LPIPS ↓ | PSNR ↑ |
> |:---|:---|:---|:---|
> | Human Face | 2.58 | 0.11 | 22.77 |
> | Architecture | 2.57 | 0.11 | 22.81 |
> | Landscape | 2.57 | 0.11 | 22.79 |
>
> *Table 1: Ablation of calibration data diversity and content*
>
> > **Q2:** Additionally, I am curious whether the experimental conclusions are consistent with different amounts of calibration data. ScaleKV only tested on 10 prompts to determine the drafters and refiners—are there more conclusions supporting this?
>
> **A2:** Thanks for the valuable question. We tested calibration sizes from 1 to 128 prompts using GPT-4o generated data. Table 2 demonstrates stable performance regardless of calibration size, validating that minimal data suffices to identify layer roles due to fundamental attention pattern differences and cache requirements.
>
> | Size | FID ↓ | LPIPS ↓ | PSNR ↑ |
> |:---|:---|:---|:---|
> | 1 | 2.58 | 0.11 | 22.78 |
> | 4 | 2.57 | 0.11 | 22.79 |
> | 16 | 2.57 | 0.11 | 22.81 |
> | 64 | 2.57 | 0.11 | 22.81 |
> | 128 | 2.57 | 0.11 | 22.81 |
>
> *Table 2: Ablation of calibration data size*
>
> > **Q3:** Regarding actual running speed rather than computational load or GPU usage, I noticed in Fig7(b) that the latency for 1024×1024 resolution is only 52ms vs. 71ms compared to full cache. Are there any optimization potentials or observations here?
>
> **A3:** Thanks for the valuable observation. ScaleKV prioritizes memory efficiency over raw speed, as VAR's main bottleneck is exponential KV cache growth at high resolutions. VAR is already significantly faster than diffusion transformers or traditional AR models of comparable size. Nevertheless, our compression method can be integrated with FlashAttention [1] to further optimize inference speed, achieving combined 2.66× speedup (77ms → 29ms) on Infinity-8B with NVIDIA A100-SXM4-80GB.
>
> | Method | Budget | Latency (ms) |
> |:---|:---|:---|
> | Full KV | 100% | 77 |
> | ScaleKV | 10% | 46 |
> | ScaleKV + FlashAttention | 10% | 29 |
>
> *Table 3: Inference latency of Infinity-8B*
>
> [1] FlashAttention: Fast and Memory-Efficient Exact Attention with IO-Awareness
>
> > **Q4:** What other drawbacks does the current KV cache method have?
>
> **A4:** Thanks for raising this important concern. As a post-training compression framework, ScaleKV's output quality is fundamentally bounded by the original VAR model's capabilities. Therefore, if the baseline quality of the original VAR models is unsatisfactory, achieving high-quality results with our method could be challenging.

---

> > ### Comment · Reviewer_74F6 · 2025-08-05
> >
> > Thank you for the supplementary experiments. I also believe that optimizing the kv cache on the var architecture is a quite good motivation. I will raise the score to "accept".

---

> > > ### Author Response · Authors · 2025-08-05
> > >
> > > Thank you for reviewing our paper and providing valuable feedback. We're glad our rebuttal addressed your concerns, and we'll include the suggested experiments in the revised manuscript. Thanks again for your time and effort.

---

### Decision · Program_Chairs · 2025-09-17

**Decision:**

Accept (poster)

**Comment:**

The paper presents ScaleKV, a scale-aware KV cache compression framework for visual autoregressive (VAR) models that reduces memory usage by up to 90% while maintaining generation quality. Its main strengths lie in the clear identification of drafter/refiner layer patterns, extensive experiments on Infinity-2B/8B models, and robustness analyses showing stable performance across calibration sizes, domains, and architectural variations. While reviewers raised concerns about generalization to non-VAR frameworks and practical impact beyond Infinity-style models, the rebuttal addressed these points with additional experiments and discussions, leading to consensus that the work merits acceptance.